# Monostotic Fibrous Dysplasia Mimicking Metastasis in the Femoral Neck on Bone Scintigraphy and ^18^F-FDG PET/CT

**DOI:** 10.3390/diagnostics10090682

**Published:** 2020-09-10

**Authors:** Wei-Liang Hung, Hung-Yen Chan, Ni-Chun Kuo, Hung-Pin Chan

**Affiliations:** 1Nephrology Division, Department of Internal Medicine, Tri-Service General Hospital, Taipei 10086, Taiwan; lydership@gmail.com; 2Nephrology Division, Department of Internal Medicine, Zouying Branch of Kaohsiung Armed Forces General Hospital, Kaohsiung 81342, Taiwan; 3Department of Nuclear Medicine, Kaohsiung Veterans General Hospital, Kaohsiung 81362, Taiwan; hongyenchan0407@yahoo.com.tw (H.-Y.C.); as123as41as@gmail.com (N.-C.K.)

**Keywords:** bone scintigraphy, fibrous dysplasia, metastasis, PET/CT

## Abstract

A 51-year-old woman who had lung adenocarcinoma was found to have a high uptake area over the right femoral neck by means of bone scintigraphy, suggesting a suspicious bony metastasis. ^18^F-FDG PET/CT was arranged, and showed an FDG-avid lesion in the same region. However, after augmented CT, a well-defined ground-glass lesion with circular calcification was found, which is the favored benign lesion of fibrous dysplasia. The following imaging of bone scintigraphy and ^18^F-FDG PET/CT presented no apparent change. This case demonstrates that the treatment scheme should not be solely guided by abnormalities in scintigraphy. Additional imaging is recommended for accurate staging or development of an appropriate treatment plan.

**Figure 1 diagnostics-10-00682-f001:**
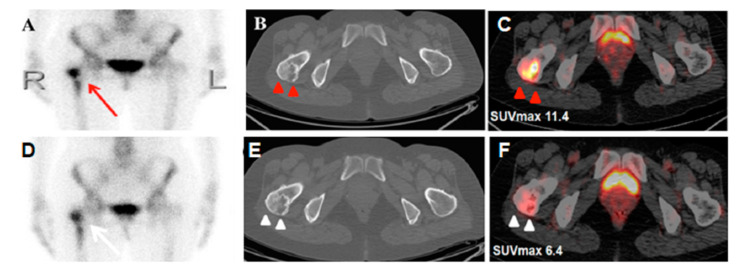
A 51-year-old woman who had lung adenocarcinoma in the right middle lobe (pT2aN0) underwent ^99m^technetium-labelled methylene diphosphonate (MDP) bone scintigraphy before an operation due to right hip soreness. She denied trauma or any operation history. Whole-body planar imaging revealed unevenly increased activity area over the right proximal femur (**A**; red arrow), suggesting a suspicious bony metastasis. Additional ^18^F-fluorodeoxyglucose positron emission tomography/computed tomography (^18^F-FDG PET/CT) showed an FDG-avid spot in the same region (SUVmax: 11.4, **C**; red arrow head). However, after augmenting the images from an axial CT, a mixed density, well-defined ground-glass lesion with cystic calcification was found in the right femoral neck (**B**; red arrow head). All these radiological features displayed a typical picture of monostotic fibrous dysplasia (FD) [1], a favored benign entity. Due to suspicious tumor recurrence one year after right lung segmentectomy, repeat FDG PET/CT was performed again. It showed a similar morphology of the FDG-avid lesion and axial CT appearance (**E**; white arrow head) in the right femoral neck, but decreased FDG metabolism as compared with the previous study (SUVmax: 6.4, **F**; white arrow head). A post-therapy bone scan presented a similar MDP-avid pattern in the right proximal femur, as compared with the prior bone scan (**D**; white arrow). FD is a dysplastic disease of the bone-forming mesenchymal cells, resulting in the development of abnormal fibrous tissue in the place of normal bone. Although it may occur as a malignant transformation in rare situations [2,3], the condition per se is usually benign. Generally, conservative management is adequate for FD patients, and education around fall prevention may minimize the risk of a femoral fracture. Nevertheless, the radiologic features of fibrous dysplasia in cancer patients could lead to a misperception in the assessment of tumor staging, and there have been several reports of FD mimicking malignancy radiologically [4,5,6,7]. Furthermore, there is no reliable threshold for FDG avidity to differentiate FD from malignancy currently, because of overlapping SUVmax values [8]. Since misidentifying terminal cases from those who are actually treatable is disastrous in cancer patients, our case demonstrates that the treatment scheme should not be solely guided by abnormalities in scintigraphy. Thus, complement examinations recommended by the National Comprehensive Cancer Network, with/without scintigraphy, are crucial for accurate staging and directing of an appropriate treatment plan.

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
