# Peer review of "Monostotic Fibrous Dysplasia Mimicking Metastasis in the Femoral Neck on Bone Scintigraphy and 18F-FDG PET/CT"

_diagnostics, 2020, doi:10.3390/diagnostics10090682_

Round 1
Reviewer 1 Report
Thank you for reminding me of this rare phenomenon. I have recommended editing by a native English speaker; otherwise it was fascinating. Would love to see a future review article on this condition.
Author Response
Point 1: Thank you for reminding me of this rare phenomenon. I have recommended editing by a native English speaker; otherwise it was fascinating. Would love to see a future review article on this condition.
Response 1:
Dear Reviewer
Thank you for your comments and suggestions. Our manuscript was already under English edition by MDPI service. English edition certificate will be provided and ready.
Sincerely
Dr Chan

Reviewer 2 Report
Dear Authors,
the manuscript show an intriguing report of bone scan and 18F FDG PET findings.
The only improvement I can suggest is to include additional headers in Figure 1 (i.e. a,b,c fo baseline and d,e,f for post-therapy images) and to provide an anterior view of follow up image of the bon scan (posterior view is reported and non clearly comparable).
Author Response
Point 1: Dear Authors, the manuscript show an intriguing report of bone scan and 18F FDG PET findings.The only improvement I can suggest is to include additional headers in Figure 1 (i.e. a,b,c for baseline and d,e,f for post-therapy images) and to provide an anterior view of follow up image of the bone scan (posterior view is reported and non clearly comparable)..
Response 1:
Dear Reviewer, We adjusted our Figure by reviewer’s suggestion and comments. Figure-1- a, b, c will be baseline images and Figure-1- d, e, f will be post-therapy images. Figure1-c was added by anterior view of bone scan after therapy. Some change in articles (see red highlight words in manuscript):
Line 26: Add new adjusted figure.
Line 32: change: (SUVmax: 11.4, Figure C)
Line 34: Add: (Figure B)
Line 38: change: previous study (SUVmax: 6.4, Figure F)
Line 38-39: Add: A post-therapy bone scan presented a similar MDP-avid pattern in the right proximal femur, as compared with the prior bone scan (Figure D).
Sincerely
Dr Chan
